

# Poor sleep quality and suicidal ideation among pregnant women during COVID-19 in Ethiopia: systematic review and meta-analysis

Aragaw Asfaw Hasen[1], Abubeker Alebachew Seid[2] and Ahmed Adem Mohammed[2]

[1] Department of Statistics, College of Natural and Computational Sciences, Samara University, Semera, Afar, Ethiopia
[2] Department of Nursing, College of Medicine and Health Sciences, Samara University, Semera, Afar, Ethiopia

Corresponding author
Aragaw Asfaw Hasen,
aragawasfaw5@gmail.com

## ABSTRACT

**Introduction.** COVID-19 has profoundly impacted the mental health and well-being of pregnant women worldwide. In Ethiopia, the poor sleep quality and suicidal ideation among pregnant women has increased due to the COVID-19 pandemic. This study aimed to provide comprehensive evidence on the prevalence and associated factors of poor sleep quality and suicidal ideation among pregnant women during COVID-19 in Ethiopia.

**Materials and Methods.** This study is based on the Preferred Reporting Items for Systematic Reviews and Meta-Analyses (PRISMA) recommendation. Data were searched from PubMed, Google Scholar, and African Journals Online from the occurrence of the COVID-19 pandemic to February 2023. Two researchers extracted the data and performed the methodological quality assessment independently. Random-effect model was used to estimate the pooled effect size and $I^2$ was used to check heterogeneity. Stata 14.0 (StataCorp, Collage Station, Texas, USA) was used for statistical analysis.

**Results.** From six studies the pooled prevalence of poor sleep quality was 55% (95% CI [0.42–0.69], $I^2 = 98.1\%$, $p < 0.001$). Age $\geq 30$ years pooled AOR = 1.95, 95% CI (0.85, 3.06), 3rd trimester pooled AOR = 3.20, 95% CI (1.82, 4.58), substance use pooled AOR = 2.51, 95% CI (0.99, 4.04), depression pooled AOR = 2.97, 95% CI (0.92, 5.02) and stress pooled AOR 2.14, 95% CI (0.24, 4.03) were associated factors of poor sleep quality. Three studies reported about suicidal ideation and pooled prevalence was 11% (95% CI: 0.09, 0.13, $I^2 = 48.2\%$, $p = 0.145$). Depression pooled AOR = 3.19, 95% CI (1.68, 4.71) was the only associated factor of suicidal ideation.

**Conclusion.** Due to COVID-19 pregnant women in Ethiopia were affected by poor sleep quality and suicidal ideation. Thus, suitable and well designed programs proposing awareness of COVID-19, mental health counseling and involvement should be designed to improve the general mental health of pregnant women.

**Trial registration.** PROSPERO registration number CRD42023389896.

## INTRODUCTION

Pregnancy is an event in a woman's life and can lead to psychological disorders. Several risk factors (such as disasters and pandemics) have been correlated with prevalence of mental disorders during pregnancy (*Keskin, Keskin & Bostan, 2022*). The COVID-19 pandemic has profoundly impacted mental health and well-being around the globe. Public health measures to control the virus's rapid spread, such as physical distancing, social isolation, lockdown, restricted movements, and quarantine, triggered fear in the general population (*Ajayi et al., 2021*). The COVID-19 pandemic has resulted in elevated mental health problems for pregnant women (*Khoury et al., 2021*; *Zeng et al., 2020*).

Pregnant women are vulnerable and suffered great psychological problems (*Wu et al., 2021*). Pregnant women's mental health is significantly impacted by their socioeconomic status, educational attainment, and social support (*Campos-garz, Riquelme-gallego & De Torre-luque, 2021*). Women who are pregnant, postpartum, miscarrying, or experiencing intimate partner violence are at high risk for developing mental health problems during the pandemic. Proactive outreach to these groups of women and enhancement of social supports could lead to prevention, early detection, and prompt treatment (*Almeida, 2020*).

Sleep is an important biological function in humans. Physical body dynamics during pregnancy lead to the deterioration of both sleep quality and sleep duration. According to the Pittsburgh Sleep Quality Index (PSQI) a score of above 5 is considered poor sleep quality (*Smyka et al., 2021*). On the other hand, suicide is a fatal action of terminating one's own life. Suicidal ideation (SI) is a thought about one's serving as an agent to kill him/herself and it is a predictor of suicide attempts and completions (*Jacobs et al., 2010*). During COVID-19 in Bangladesh about 8.2% of young adults reported having at least suicidal thoughts from the pandemic's inception to survey time (*Mamun et al., 2021*). Suicidal ideation (SI) is measured by the World Health Organization (WHO) composite international diagnostic interview (CIDI) (*Kessler & Üstün, 1981*).

The COVID-19 pandemic was expected to have terrible effects on mothers' mental health in Africa because of the continent's poor health systems, inadequate mental health policies, and unreliable maternal care (*Ajayi et al., 2021*; *Senkyire et al., 2023*). Studies showed that during the COVID-19 pandemic the prevalence of mental disorders among pregnant women was very high. For instance, of all participating pregnant women, 88% had poor sleep quality during the COVID-19 pandemic (*Alan et al., 2020*). Similarly, the prevalence of poor sleep quality among pregnant women during the pandemic was high. Maternal age, third trimester pregnancy, being multigravida, and having comorbidity were among associated factors of poor sleep quality of pregnant women during COVID-19 (*Amare, Chekol & Aemro, 2022*). The prevalence of poor sleep quality was 71.3% (*Bogale et al., 2022*) and 30.8% (*Anbesaw et al., 2021a*; *Anbesaw et al., 2021b*).

In addition, suicidal ideation is communal in the third trimester, particularly in pregnant women with poor marital satisfaction, high attachment anxiety, prenatal depression and prenatal anxiety (*Zhang et al., 2022*). The prevalence of suicidal ideation among pregnant women was 11.8% (*Belete et al., 2021*) and 13.3% of women who are on antenatal care were with suicidal ideation (*Anbesaw et al., 2021b*). Women who are on antenatal care developed

suicidal ideation due to different factors including, absence of cohabiting partners, history of abortion, depression, anxiety, intimate partner violence, poor sleep quality and stress (*Anbesaw et al., 2021b*). There are single level study findings in Ethiopia (*Amare, Chekol & Aemro, 2022*; *Anbesaw et al., 2021a*; *Bogale et al., 2022*; *Dule et al., 2021*; *Jemere et al., 2021*; *Belete et al., 2021*; *Molla, Nigussie & Girma, 2022*; *Takelle, Muluneh & Biresaw, 2022*; *Anbesaw et al., 2021a*) on the prevalence of poor sleep quality and suicidal ideation among pregnant women during COVID-19. The reports on prevalence of poor sleep quality and suicidal ideation in Ethiopia were high, and the results were inconsistent.

Therefore, a thorough investigation is necessary, as is the provision of thorough evidence regarding the prevalence and contributing variables of poor sleep quality and suicidal ideation among pregnant women in Ethiopia. No study reported in a summarized way. By measuring the combined prevalence of poor sleep quality, suicidal ideation, and their associated factors, this study intends to close this gap. This study offers proof of mental health issues in pregnant women to academics and decision-makers.

## OBJECTIVES

### General objective
- This study aimed to provide comprehensive evidence on the prevalence and associated factors of poor sleep quality and suicidal ideation among pregnant women during COVID-19 in Ethiopia.

### Specific objectives
- To estimate the pooled prevalence of poor sleep quality and suicidal ideation during COVID-19 among pregnant women in Ethiopia.
- To review associated factors of poor sleep quality and suicidal ideation during the COVID-19 pandemic among pregnant women in Ethiopia.

## MATERIALS AND METHODS

### Protocol registration
The Preferred Reporting Items for Systematic Reviews and Meta-Analyses (PRISMA) 2020 recommendation was used (*Moher et al., 2015*). The protocol for this systematic review and meta analysis was registered by International Prospective Register of Systematic Reviews with PROSPERO registration number: CRD42023389896.

### Search strategy
Searching of literature was done by two authors (AAH and AAS) individually. Disagreement was handled by discussion with third author (AAM). Databases PubMed, African Journals Online and Google Scholar, and papers published from the occurrence of the COVID-19 pandemic to February 2023 were searched. Observational studies that evaluated sleep quality and suicidal ideation during COVID-19 among pregnant women in Ethiopia were considered. Systematic searches were carried out by combining every possible predefined search term determined by Medical Subject Headings (MeSH) and keywords. The following key words were used: "COVID-19", "2019 novel coronavirus disease", "2019 novel
coronavirus infection",“2019 ncov disease", "2019 ncov infection", "covid 19 pandemic", "covid 19 pandemics", "covid 19 virus disease", "covid 19 virus infection", "COVID19", "coronavirus disease 2019", "coronavirus disease 19", "sars coronavirus 2 infection", "sars cov 2 infection", "severe acute respiratory syndrome coronavirus 2 infection", "SARS-CoV-2", "2019 novel coronavirus", "2019 novel coronavirus", "2019- nCoV", "covid 19 virus", "covid19 virus", "Coronavirus disease 2019 virus", "SARS coronavirus 2", "SARS cov 2 virus", "severe acute respiratory syndrome coronavirus 2", "Wuhan coronavirus", "Wuhan seafood market pneumonia virus", "mental illness", "psychiatric problem", "psychology problem", "mental health effect", "psychological disturbance", "mental disorder", "psychiatric illness", "psychiatric diseases", "psychiatric disorders", "behavior disorders", "severe mental disorder", "sleeping quality", "poor sleep quality", "insomnia", "suicide", "suicidal ideations", "pregnant women", "prenatal", " perinatal", "postpartum", "antenatal", "postnatal", "puerperal", "lactating women", "reproductive age women", "child bearing women" and "Ethiopia".

Reference lists of full text articles included in the review were checked to identify any potentially eligible study for this review. Systematic procedure confirmed that the literature search encompasses all published studies on the prevalence and associated factors of poor sleep quality and suicidal ideation during COVID-19 among pregnant women in Ethiopia. From the searched results duplicates were detached by using Mendeley (*Kwon et al., 2015*). The search strategy for finding suitable studies from databases is presented in Supplemental File S1.

## Eligibility criteria
### Inclusion criteria
For this systematic review and meta-analysis, observational studies that focused on the prevalence and associated factors of poor sleep quality and suicidal ideation among pregnant women during COVID-19 in Ethiopia were considered. Moreover, the inclusion criteria is stated as:

*Setting:* Studies in Ethiopia were the main emphasis of this review.

*Population:* All pregnant women in Ethiopia.

*Study design*: Observational studies on prevalence of poor sleep quality and suicidal ideation and their associated factors.

*Language:* Only English.

## Exclusion criteria
The following types of studies were excluded: studies focusing on the whole population, studies that do not report prevalence and associated factors of poor sleep quality and suicidal ideation among pregnant women, systematic reviews and meta-analyses, randomized controlled trials, editorials, and conference abstracts and opinions.

## Outcome measures
The primary outcomes were the pooled prevalence of prevalence of poor sleep quality and suicidal ideation among pregnant women during COVID-19 in Ethiopia. The secondary

outcomes were factors associated with poor sleep quality and suicidal ideation among pregnant women during COVID-19 in Ethiopia.

## Selection of studies

Two researchers (AAH and AAS) reviewed studies based on eligibility criteria. Firstly, the authors evaluated titles and abstracts of the studies from the searched databases. Then, full-text screening was done to screen the full texts. We have provided a reasoning for inclusion and exclusion of studies in the PRISMA diagram. The list of suitable studies for this systematic review and meta-analysis were arranged.

## Data extraction

The data were collected by two researchers (AAH and AAS) individually. Pretest on the data extraction form by using two pilot surveyed studies was done. This helps ensure the collection of all necessary data for this systematic review and meta-analysis is accurate. Disagreements were handled by deep discussion among the investigators. All relevant data were collected from the original studies. Specifically, author's last name, year of publication, region, study design, number of cases, sample size, sampling design, instrument, study population, average age and prevalence of poor sleep quality and suicidal ideation with their associated factors were extracted.

## Methodological quality assessment

Two researchers (AAH and AAM) separately evaluated the quality of the included studies by the Newcastle-Ottawa Scale (NOS) (*Peterson, Welch & Losos, 2011*). According to this scale, three parameters named selection, comparability, and assessment of exposure/outcome were considered. Studies with less than 5 scores were considered low quality, 5-7 scores of moderate quality, and more than 7 scores of high quality (*Ssentongo et al., 2020*). Studies with moderate and above quality score were considered for this study. Disagreements on scores between reviewers were resolved by discussion. The final quality score was presented in Table 1.

## Data synthesis

The statistical data analysis was done by Stata 14.0 (StataCorp, Collage Station, Texas, USA). The pooled prevalence with 95% confidence interval (CI) by using random-effect model with the generic inverse variance method was computed. Evaluation of heterogeneity was checked by $I^2$ and Cochran's Q-statistics (*Bowden et al., 2011*; *Zhu et al., 2020*). Subgroup analyses was performed by regions, year of publication and sampling methods used in individual studies to decide the source of heterogeneity. Publication bias was checked by Doi plot and the Luis Furuya Kanamori (LFK) index was applicable when the number of studies was small and used to assess asymmetry (*Furuya-Kanamori, Barendregt & Doi, 2018*) (*Furuya-Kanamori & Doi, 2021*). The LFK index value out of the interval -1 and 1 were considered asymmetry (existence of publication bias) (*Furuya-Kanamori & Doi, 2021*).

**Table 1** Study characteristics and quality of the included studies for poor sleep quality and suicidal ideation among pregnant women during the COVID-19 pandemic in Ethiopia.

| No | Authors | Year | Region | Mental disorders | Population | Study design | Sampling method | Sample size (n) | Cases | P (%) | Instrument | Age (average) | Quality |
|---|---|---|---|---|---|---|---|---|---|---|---|---|---|
| 1 | Dule et al. (2021) | 2021 | Oromiya | Poor sleep quality | All PW | CS | Census/consecutive | 228 | 130 | 57 | PSQI | 30.79 | 7 |
| 2 | Jemere et al. (2021) | 2021 | Amhara | Poor sleep quality | All PW | CS | Systematic | 411 | 281 | 68.4 | PSQI | 25.7 | 8 |
| 3 | Bogale et al. (2022) | 2022 | SWE | Poor sleep quality | RAW | CS | Multistage | 606 | 432 | 71.3 | PSQI | 25 | 8 |
| 4 | Anbesaw et al. (2021b) | 2021 | Oromiya | Poor sleep quality | All PW | CS | Systematic | 415 | 128 | 30.8 | PSQI | 25.22 | 8 |
| 5 | Takelle, Muluneh & Biresaw (2022) | 2022 | Amhara | Poor sleep quality | All PW ≥18 years | CS | Systematic | 415 | 175 | 42.2 | PSQI | 28 | 8 |
| 6 | Amare, Chekol & Aemro (2022) | 2022 | Amhara | Poor sleep quality | All PW | CS | Systematic | 423 | 265 | 62.8 | PSQI | 28 | 8 |
| 7 | Molla, Nigussie & Girma (2022) | 2022 | SNNP | Suicidal ideation | All PW | CS | Multistage | 504 | 47 | 9.3 | SBQ-R | 27.56 | 8 |
| 8 | Anbesaw et al. (2021a) | 2021 | Oromiya | Suicidal ideation | All PW | CS | Systematic | 415 | 55 | 13.3 | CIDI | 25.22 | 8 |
| 9 | Belete et al. (2021) | 2021 | SNNP | Suicidal ideation | All PW | CS | Systematic | 738 | 87 | 11.3 | CIDI | 25.54 | 8 |

**Notes.**

CIDI, Composit International Diagnostic Interview; SBQ-R, *Suicide Behaviors Questionnaire-Revised*; PSQI, Pittsburgh Sleep Quality Index; SNNP, Southern Nations Nationalities and People; SWE, South West Ethiopia; CS, Cross-sectional; PW, Pregnant women; P, Prevalence; RAW, Reproductive Age Women; AOR, adjusted odds ratio; CI, Confidence Interval; IPV, Intimate Partner Violence.

# RESULTS

A PRISMA diagram demonstrating steps of the database search and the refining process for the study on poor sleep quality and suicidal ideation among pregnant women during the COVID-19 pandemic was shown in Fig. 1. From our databases search primarily 22 studies were identified. Due to duplication four studies were removed. About 18 full text studies were examined and nine studies were removed by reasons that did not encountered inclusion criteria. Finally, we recognized nine studies appropriate to this systematic review and meta-analysis.

## Study characteristics

In this systematic review and meta-analysis, we included nine cross sectional studies (*Amare, Chekol & Aemro, 2022*; *Anbesaw et al., 2021a*; *Anbesaw et al., 2021b*; *Bogale et al., 2022*; *Dule et al., 2021*; *Jemere et al., 2021*; *Belete et al., 2021*; *Molla, Nigussie & Girma, 2022*; *Takelle, Muluneh & Biresaw, 2022*; *Anbesaw et al., 2021b*) on the prevalence and associated factors of poor sleep quality and suicidal ideation among pregnant women during the COVID-19 pandemic in Ethiopia. Based on the types of mental disorders; six studies (*Amare, Chekol & Aemro, 2022*; *Bogale et al., 2022*; *Anbesaw et al., 2021a*; *Anbesaw et al., 2021b*; *Jemere et al., 2021*; *Takelle, Muluneh & Biresaw, 2022*; *Dule et al., 2021*) were reported about poor sleep quality; three studies (*Belete et al., 2021*; *Anbesaw et al., 2021b*; *Molla, Nigussie & Girma, 2022*) were reported about suicidal ideation. Furthermore, the summarized data of the key characteristics of the included studies for this review were showed in Table 1.

## Quality of included studies

The quality of included studies using the modified Newcastle Ottawa scale (NOS) for cross-sectional studies quality assessment was presented in Table 1. We considered moderate quality, and high-quality studies for this review. Thus, one study (*Dule et al., 2021*) that was appraised as moderate quality and eight studies (*Amare, Chekol & Aemro, 2022*; *Bogale et al., 2022*; *Anbesaw et al., 2021b*; *Belete et al., 2021*; *Jemere et al., 2021*; *Molla, Nigussie &*

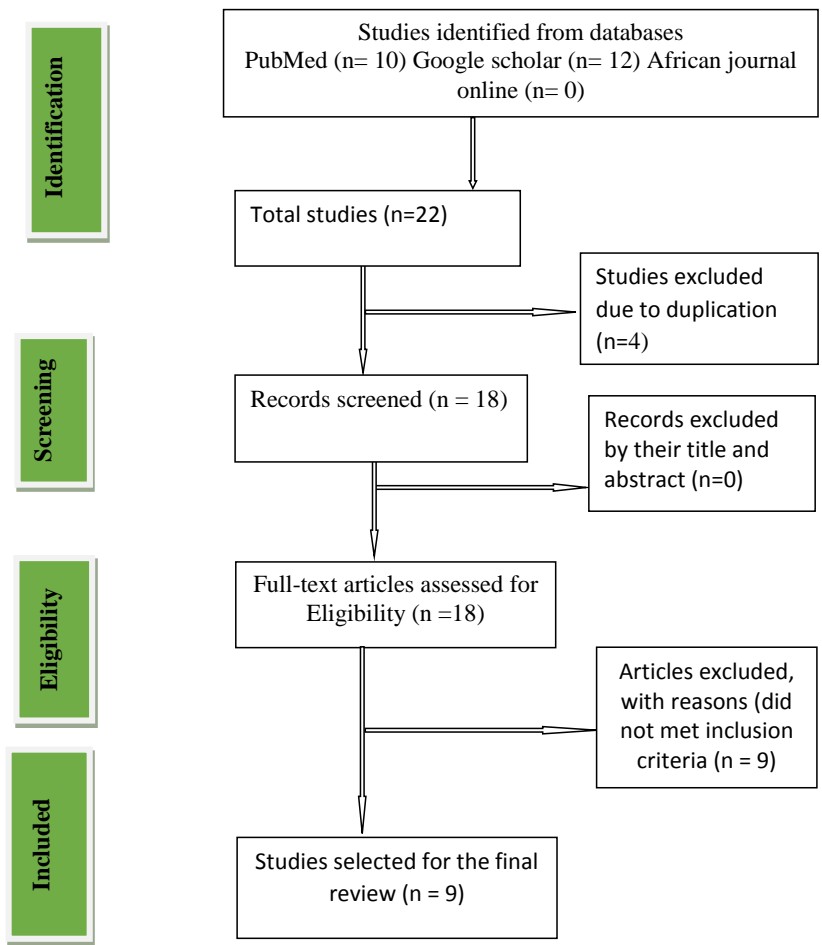

**Figure 1** Preferred reporting items for systematic reviews and meta-analyses (PRISMA) flow chart.

*Girma, 2022*; *Takelle, Muluneh & Biresaw, 2022*) that were appraised as high quality were considered for final systematic review and meta analysis.

## Publication bias

Publication bias was evaluated by using Doi plot (*Furuya-Kanamori, Barendregt & Doi, 2018*), a method used to see asymmetry and by the LFK index (*Furuya-Kanamori & Doi, 2021*), used to detect and quantify asymmetry of study effects. For poor sleep quality studies (LFK index $= -1.49$, Egger's test $p$-value $=0.531$), *i.e.*, there is minor asymmetry, but the Egger's test result is not statistically significant with $p$-value $=0.531$, indicating that no significant influence was found on the pooled result and therefore no publication bias. For suicidal ideation studies (LFK index $= -0.05$, Egger's test $p$-value $= 0.674$). *i.e*, both LFK index and Egger's test supports no publication bias (Fig. 2).

## Pooled prevalence of poor sleep quality

Six studies reported the prevalence of anxiety, and the pooled prevalence of poor sleep quality was found to be 55% (95% CI [0.42–0.69], $I^2 = 98.1\%$, $p < 0.001$). As shown in

A.

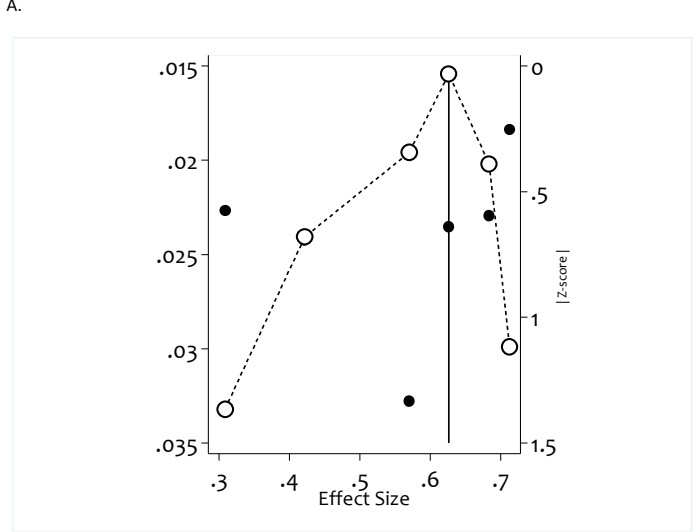

B.

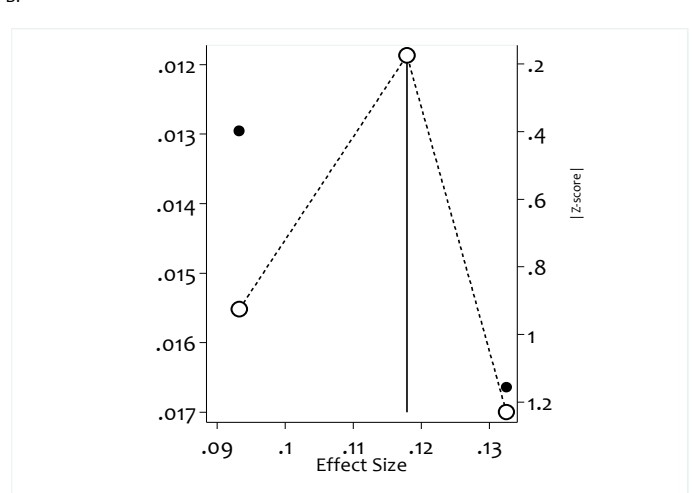

**Figure 2** **Assessment of publication bias of included studies using Doi plot, LFK index and Egger's test for included studies.** (A) Doi plot and LFK index for poor sleep quality studies, LFK index $= -1.49$; Egger's test (p_value) $= .531$. (B) Doi plot and LFK index for suicidal ideation studies; LFK index $= -.05$; Egger's test (p_value) $= .674$.

Fig. 3 there is substantial heterogeneity among study findings on prevalence of poor sleep quality among pregnant women during the COVID-19 pandemic.

## Subgroup analysis of poor sleep quality by region

From the subgroup analysis of prevalence of poor sleep quality by region, the pooled prevalence of poor sleep quality in Oromiya, Amhara and South West Ethiopia was 44%, 58% and 71% respectively (Fig. 4). The prevalence was higher in the South West Ethiopia region compared to the others. There was statistically significant heterogeneity among regions ($Q = 6.39$, $df = 2$, $p = 0.032$).

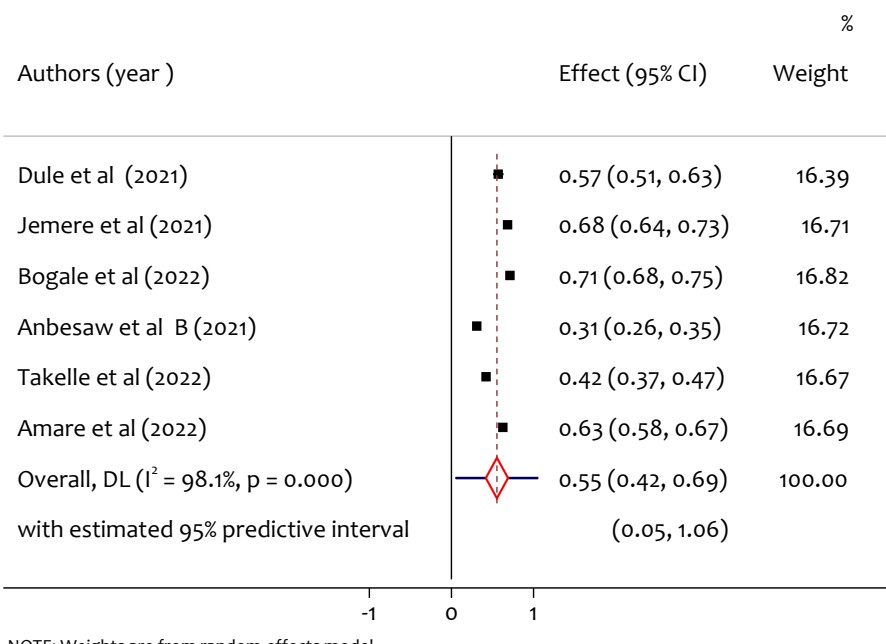

**Figure 3** A forest plot for the prevalence of poor sleep quality among pregnant women during the COVID-19 pandemic.

## Subgroup analysis of poor sleep quality by year of publication

From the subgroup analysis by publication year (Fig. 5), the pooled prevalence of poor sleep quality in 2021 and 2022 was 52% and 59%, respectively. The prevalence of poor sleep quality among pregnant women was high in studies published in 2022. The heterogeneity test between group result ($Q = 0.2$, $df = 1$, $p = 0.657$) implies that there was no significant heterogeneity in years of publication on the prevalence of poor sleep quality.

## Subgroup analysis of poor sleep quality by sampling method

From the subgroup analysis by sampling method (Fig. 6) the pooled prevalence of poor sleep quality in study using sampling methods census was 57%, systematic sampling was 63% and multistage sampling was 71%. The heterogeneity test between group ($Q = 20.60$, $df = 3$, $p = 0.000$) implies that there is statistically significant heterogeneity among sampling methods.

## Pooled prevalence of suicidal ideation

From three studies, the pooled prevalence of suicidal ideation was 11% (95% CI [0.09–0.13], $I^2 = 48.2\%$, $p = 0.145$) as shown in Fig. 7. No significant heterogeneity was observed. We did not do subgroup analysis to assess the source of heterogeneity.

## Associated factors and pooled effect size

As shown in Table 2, the pooled adjusted odds ratio of age >= 30 years pregnant women was 1.95, 95% CI (0.85, 3.06), suggesting that the odds of developing poor sleep quality

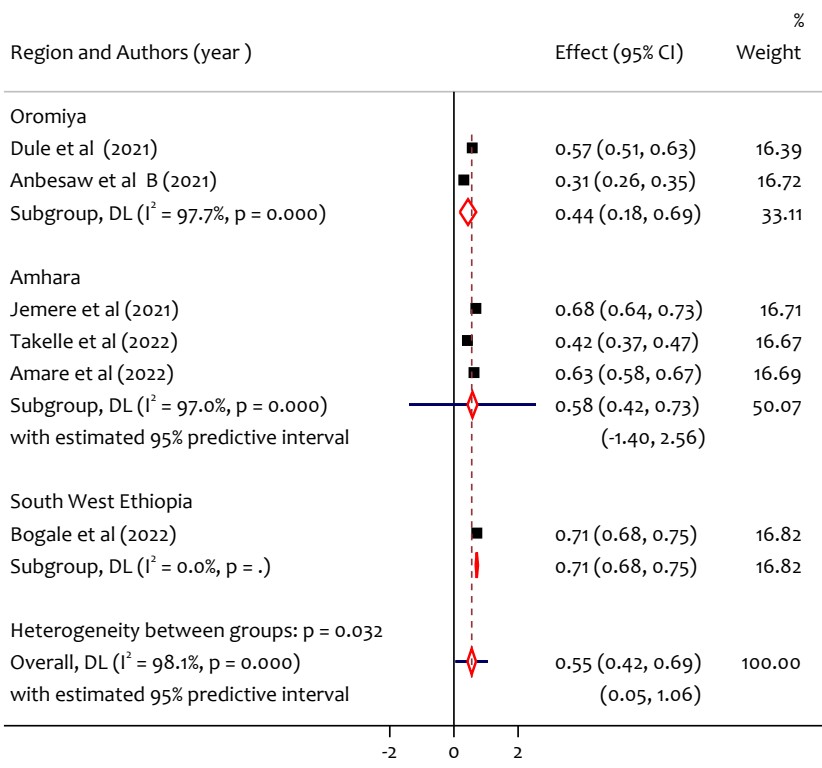

NOTE: Weights and between-subgroup heterogeneity test are from random-effects model

**Figure 4** Subgroup analysis of prevalence of poor sleep quality among pregnant women during the COVID-19 pandemic by region.

among pregnant women was 1.95 times more with age >= 30 years pregnant women than women with age <30 years. The pooled adjusted odds ratio of 3rd trimester pregnant women was 3.20, 95% CI (1.82, 4.58), implying that the odds of developing poor sleep quality among pregnant women on 3rd trimester gestational age was 3.20 times more than women in the 2nd or 1st trimester.

The pooled adjusted odds ratio of substance use among pregnant women was 2.51, 95% CI (0.99, 4.04), suggesting that the odds of developing poor sleep quality among pregnant women with substance use was 2.51 times more than women without substance use. The pooled adjusted odds ratio of pregnant women with depression was 2.97, 95% CI (0.92, 5.02) suggests that the odds of developing poor sleep quality among pregnant women with depression was 2.97 times more than women with out depression. Likewise, the pooled adjusted odds ratio of pregnant women with stress was 2.14, 95% CI (0.24, 4.03) suggests that the odds of developing poor sleep quality among pregnant women with stress was 2.14 times more than women without stress.

We also reported pooled effect size of associated factors of suicidal ideation among pregnant women. The pooled adjusted odds ratio of pregnant women with depression was 3.19, 95% CI (1.68, 4.71) suggesting that the odds of developing suicidal ideation among pregnant women with depression was 3.19 times more than women that do not

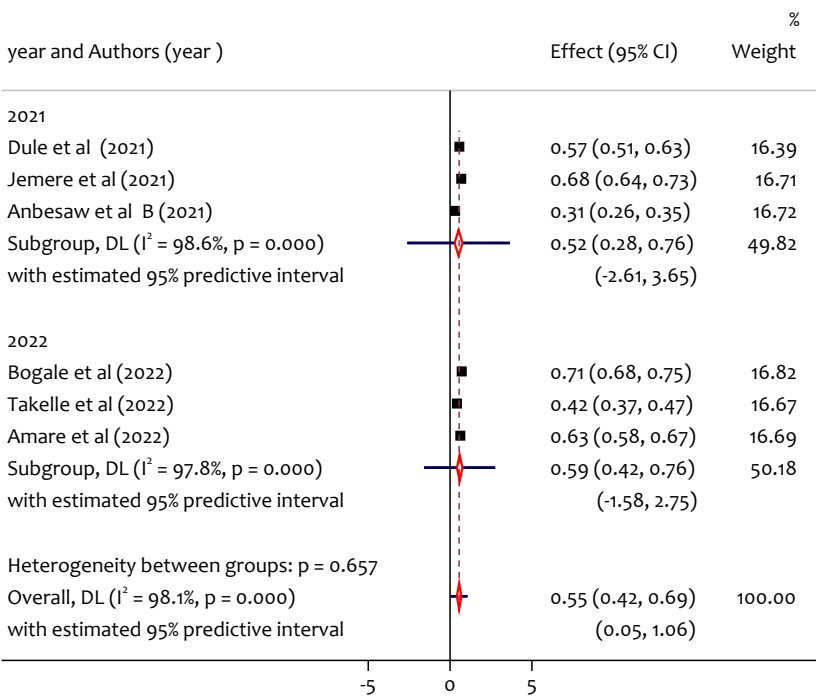

**Figure 5** Subgroup analysis of prevalence of poor sleep quality among pregnant women during the COVID-19 pandemic by year of publication.

have depression. The pooled adjusted odds ratio of pregnant women with intimate partner violence was 3.83, 95% CI (−0.67, 8.34) suggesting that the odds of developing suicidal ideation among pregnant women with intimate partner violence was 3.83 times more than women without intimate partner violence. Due to the limited number of studies available, pooling the adjusted odds ratio for the associated factors of poor sleep quality and suicidal ideation is not possible. Hence, we summarized the collected findings in Table 3.

## DISCUSSION

In this comprehensive study we have shown that due to the COVID-19 pandemic poor sleep quality and suicidal ideation increased among pregnant women in Ethiopia. This study is the first of its kind that assessed the pooled prevalence and associated factors of poor sleep quality and suicidal ideation among pregnant women during COVID-19 in Ethiopia. This study includes nine papers. We supposed that all of the included studies are conducted with in ethical guidelines. The pooled prevalence of poor sleep quality and suicidal ideation with their associated factors were discussed.

The pooled prevalence of poor sleep quality was found to be 55%. This is higher than the study in Changsha, China where the prevalence was 37.65% (*Cai et al., 2022*), the study in Wuhan, China where the prevalence was 37.6% (*Xu et al., 2021*), the study in Bangladesh where the prevalence was 38.88% (*Shaun et al., 2022*) and lower than worldwide meta analysis result where the prevalence was 49% (*Yan, Ding & Guo, 2020*). From the subgroup

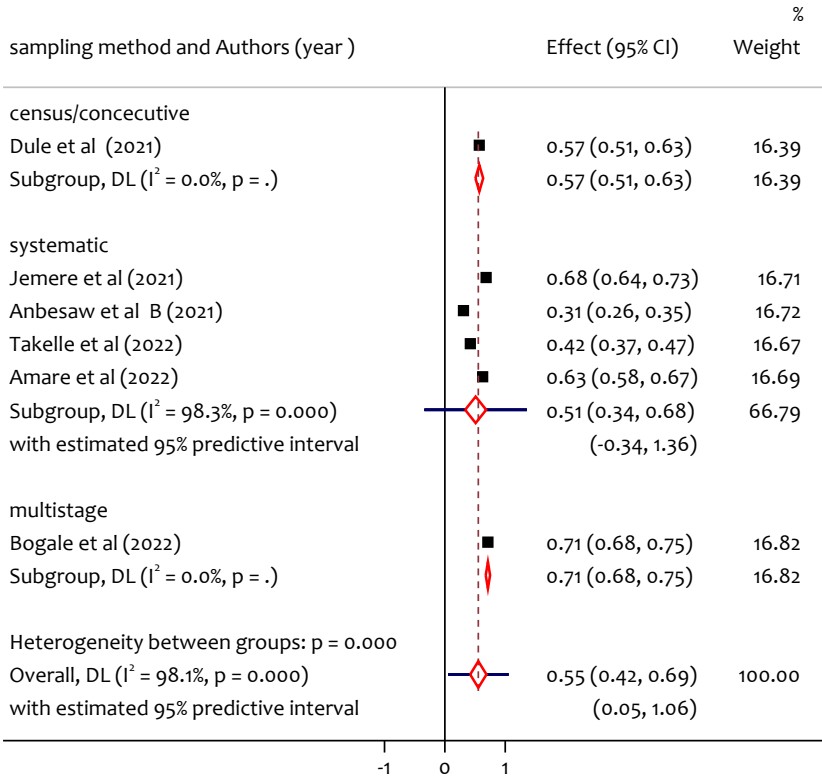

**Figure 6** Subgroup analysis of prevalence of poor sleep quality among pregnant women during the COVID-19 pandemic by sampling method.

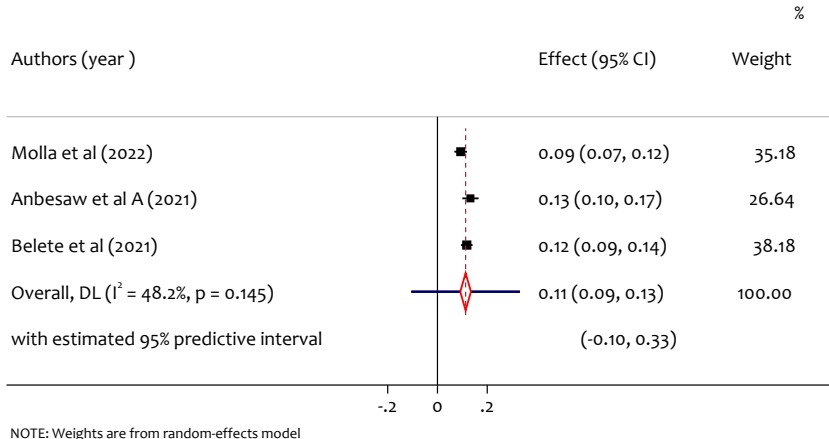

**Figure 7** A forest plot for the prevalence of suicidal ideation among pregnant women during the COVID-19 pandemic.

**Table 2  Pooled AOR for associated factors of poor sleep quality and suicidal ideation among pregnant women during the COVID-19 pandemic in Ethiopia.**

| Mental illness | Numbers of studies | Variables | Reference category | Pooled AOR ((95% CI) | Heterogeneity | |
|---|---|---|---|---|---|---|
| | | | | | I$^2$(%) | p-value |
| Poor sleep quality | 4 | Age ≥ 30 | <30 | 1.95 (0.85, 3.06) | 0.00 | 0.616 |
| >> | 3 | Gestational age in 3rd trimester | 2nd or 1st trimester | 3.20 (1.82, 4.58) | 0.00 | 0.405 |
| >> | 2 | Substanse use | Not | 2.51(0.99, 4.04) | 0.00 | 0.455 |
| >> | 2 | Depression | Not | 2.97(0.92, 5.02) | 60.50 | 0.110 |
| >> | 2 | Stress | Not | 2.14(0.24, 4.03) | 12.8 | 0.284 |
| Suicidal ideation | 2 | Depression | Not | 3.19(1.68, 4.71) | 0.00 | 0.619 |
| >> | 2 | IPV | Not | 3.83(−0.67, 8.34) | 44.8 | 0.178 |

analysis among regions, we found significant heterogeneity and prevalence of poor sleep quality by region, the pooled prevalence in Oromiya, Amhara and South West Ethiopia was 44%, 58% and 71% respectively, the prevalence was higher in the South West Ethiopia region. This might be due to the lack of awareness and poor health system in the new emerging region which requires special attention to improve maternal mental health during the pandemic. From the subgroup analysis by publication year the pooled prevalence of poor sleep quality in 2021 and 2022 was 52% and 59% respectively. The heterogeneity test between groups implies that there was no significant heterogeneity on the prevalence of poor sleep quality in publication years. On the other hand, we also observed heterogeneity on findings based on sampling methods on the prevalence of poor sleep quality among pregnant women during COVID-19 in Ethiopia. From this the prevalence of poor sleep quality in the study using sampling methods census was 57%, systematic sampling was 63% and multistage sampling was 71%. The heterogeneity might be due to the difference in the nature of the sampling methods to select samples (representatives) for making inferences in a given study.

The pooled prevalence of suicidal ideation was 11%. The result is consistent with a systematic review result prevalence worldwide that was between 2.73% to 18% (*Carolina et al., 2022*). The result is higher than the Brazilian pregnant women prevalence of suicidal ideation which was 3.9%. This might be due to the differences in the quality of maternal health service between countries. The result is in line with study findings on the general population pooled prevalence of suicidal ideation during COVID-19 which was 12.1% (*Farooq et al., 2021*) and is lower than study in Australian adults prevalence of suicidal ideation which was 17.5% (*Wiley et al., 2021*).

Our study also demonstrated the associated risk factor for poor sleep quality and suicidal ideation among pregnant women during COVID-19 in Ethiopia. For pregnant women age >= 30, the odds of developing poor sleep quality are more than with age <30 years. Gestational age in the 3rd trimester increases the odds of developing poor sleep quality among pregnant women more so women in the 2nd or 1st trimester. This finding is supported by study results from *Jelly et al. (2021)* which outlined that gestational age

**Table 3  A summary review of significant associated factors of poor sleep quality and suicidal ideation among pregnant women during COVID-19 pandemic in Ethiopia.**

| Authors (year) | Mental disorders | Associated factors | AOR (95% CI) |
|---|---|---|---|
| *Jemere et al. (2021)* | Poor sleep quality | Age of the mother age 20–30 years | 4.3 (1.8, 9.9) |
| >> | >> | Gestational age second trimester | 2.46 (1.2, 4.9) |
| >> | >> | Multiparous women | 2.1(1.24, 3.6) |
| *Bogale et al. (2022)* | Poor sleep quality | Palpable /visible thyroid gland | 2. 12 (1.08,3.82) |
| >> | >> | Having premenstrual syndrome | 1.86 (1.38, 3.12) |
| *Takelle, Muluneh & Biresaw (2022)* | Poor sleep quality | Being first trimesters | 2.31(1.16, 4.61) |
| >> | >> | Intimate Partner Violence | 5.57(2.19, 14.68) |
| *Amare, Chekol & Aemro (2022)* | Poor sleep quality | Having co-morbidity | 3.57( 1.45, 8.78) |
| >> | >> | Being multigravida | 2.72( 1.34, 5.50) |
| *Molla, Nigussie & Girma (2022)* | Suicidal ideation | Being unmarried | 5.69 (1.19, 27.23) |
| >> | >> | Gestation age greater than 27 weeks | 4.92 (1.67, 14.53) |
| >> | >> | History of having chronic medical illness | 4.47 (1.35, 14.85) |
| *Anbesaw et al. (2021a)* | Suicidal ideation | Marital status with lack of cohabiting partners | 2.80 (1.23, 6.37) |
| >> | >> | History of abortion | 2.45(1.03, 5.93) |
| >> | >> | Anxiety | 2.99 (1.24, 7.20) |
| >> | >> | Poor sleep quality | 2.85 (1.19, 6.79) |
| >> | >> | Stress | 2.50 (1.01, 5.67) |

**Notes.**
AOR, Adjusted Odds Ratio; CI, Confidence Interval.

has direct association with psychological impact scores. We also found that depression and stress increases the odds of developing poor sleep quality among pregnant women during COVID-19. Similarly, depression and stress increase the odds of developing suicidal ideation among pregnant women. Intimate partner violence also maximizes the odds of developing suicidal ideation among pregnant women during COVID-19 in Ethiopia. This is supported by study findings (*Carolina et al., 2022*).

However, this study has both strengths and weaknesses. Two researchers (AAH and AAS) independently conducted the searching, screening, data extraction, and methodological quality assessment. The quality of the included studies was evaluated using the Newcastle-Ottawa Scale. Lack of adequate number of studies on the psychological effects of COVID-19 across all regions of Ethiopia was the limitation of this study.

# CONCLUSION

Due to the COVID-19 pandemic pregnant women in Ethiopia were affected by poor sleep quality and suicidal ideation. The prevalence of poor sleep quality and suicidal ideation among pregnant women were high. Providing pregnant women with appropriate and well-designed programs that promote awareness of COVID-19, mental health counseling, and involvement to improve their general mental health is crucial. This review has helped to highlight the need of interventions that can be taken during COVID-19. Further studies could be conducted on the mental health effects of COVID-19 among pregnant women in Ethiopia.

**Abbreviations**

| | |
|---|---|
| **AOR** | Adjusted odds ratio |
| **CI** | Confidence Interval |
| **MeSH** | Medical Subject Headings |
| **NOS** | Newcastle Ottawa Quality Assessment Scale |
| **PRISMA** | Preferred Reporting Items for Systematic Review and Meta-Analysis |
| **SI** | Suicidal Ideation |

### Funding
The authors received no funding for this work.

### Competing Interests
The authors declare there are no competing interests.

### Author Contributions
- Aragaw Asfaw Hasen conceived and designed the experiments, performed the experiments, analyzed the data, prepared figures and/or tables, authored or reviewed drafts of the article, and approved the final draft.
- Abubeker Alebachew Seid conceived and designed the experiments, performed the experiments, analyzed the data, prepared figures and/or tables, authored or reviewed drafts of the article, and approved the final draft.
- Ahmed Adem Mohammed conceived and designed the experiments, performed the experiments, analyzed the data, prepared figures and/or tables, authored or reviewed drafts of the article, and approved the final draft.

### Data Availability
The data for this research is available in the Supplemental File.

### Supplemental Information
Supplemental information for this article can be found online at http://dx.doi.org/10.7717/peerj.16038#supplemental-information.

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
