# Peer review of "Poor sleep quality and suicidal ideation among pregnant women during COVID-19 in Ethiopia: systematic review and meta-analysis"

_PeerJ, doi:10.7717/peerj.16038_

## Round 0.1 · original submission · Major Revisions

I have now received the reviewers' comments on your manuscript. They have suggested some revisions to your manuscript. Therefore, I invite you to respond to the reviewers' comments and revise your manuscript.

Reviewer 1 ·

Basic reporting

The objective of the study were clearly stated. The authors provided a clear and comprehensive description of the study designs, and details of the collection process of each clinical study. The publication bias was discussed.

Experimental design

The Methodological quality assessment section need more detail. The author may want to add more detail about key words selection.

There are 4 studies were excluded due to duplication in Figure 1, these studies should not be considered at very beginning.

The population of the studies were listed as "All pregnant women in Ethiopia." The author may want to justify whether the data from the different studies are interpretable for this population.

The exclusion criteria stated that the Studies elaborate whole population were excluded. There could be pregnant women in those studies, the author should add more justification on this.

The statistical method need more information. I suggest the author add the baseline variable comparison between studies. Also, the author may want to consider propensity score method to see if there are any selection bias between studies.

Validity of the findings

In the result part, authors listed results for different subgroup analyses. As mentioned above, without baseline characteristic justification or comparison, it it hard to justify the subgroup results. The distribution for each of the selected variables should be discussed.

Also, the subgroup analyses were done separately, the author need to add discussion about the correlation between different variables.

Reviewer 2 ·

Basic reporting

Clear and unambiguous, professional English used throughout.

Experimental design

Not. The review has limitations in the search.

Validity of the findings

the poor search may compromise the validity

Additional comments

PeerJ 86474

"Data were searched from PubMed, Google Scholar, and African Journals Online" are these two data base enough? There are major data bases (embase, chinhal, ect) that could be covered. The google scholar is not good for searching as it lacks systematic searching strategies. So, the overall search is narrow.
In the pubmed, How was the search done? Are these only searches via keywords or via keywords and mesh heading? It require both.
The PICO format requires following and being described.
Details of the search strategies require submission as a supplementary file, not the main file.
The poor sleep qualities require being defined clearly, and how were it measured. As well as the suicidal ideation. Require to provide the details of the duration of the prevalence measures, such as is it were measured in last 1 year or 6 months.
The inclusion and exclusion criteria should be presented in the same manner, not one by numbering.
Three of the exclusion criteria are not clearly understandable due to the sentence are not completed.
Although the statistics say there is no significant publication bias, the graph itself is proof of the publication bias that may be possible to minimize by improving the search quality.
This article may improve the introduction, discussion, and justification of the study.
https://doi.org/10.1016/j.jadr.2021.100262
https://doi.org/10.2147/RMHP.S330282
https://doi.org/10.1186/s43045-021-00103-x
https://doi.org/10.1371/journal.pgph.0000187

---

## Round 0.2 · accepted · Accept

Many thanks for addressing all the issues.

Reviewer 1 ·

Basic reporting

The author addressed my comments. No further comments.

Experimental design

None

Validity of the findings

None

Additional comments

None

Reviewer 2 ·

Basic reporting

Clear and unambiguous, professional English used throughout.

Experimental design

Research question well defined, relevant & meaningful. It is stated how research fills an identified knowledge gap.

Validity of the findings

Conclusions are well stated, linked to original research question & limited to supporting results.